# Genetic signature of blind reintroductions of Iberian ibex (*Capra pyrenaica*) in Catalonia, Northeast Spain

Tânia Barros[1][☉]*, Joana M. Fernandes[1][☉], Eduardo Ferreira[1], João Carvalho[1,2], Marta Valldeperes[2], Santiago Lavín[2], Carlos Fonseca[1,3], Jordi Ruiz-Olmo[4], Emmanuel Serrano[2,5]

**1** Departamento de Biologia & CESAM (Centro de Estudos do Ambiente e do Mar) Universidade de Aveiro, Campus Universitário Santiago, Aveiro, Portugal, **2** Dept Medicina i Cirurgia Animals Facultat de Veterinària, Wildlife Ecology & Health group (WE&H) Servei d' Ecopatologia de Fauna (SEFaS), Universitat Autònoma de Barcelona (UAB), Bellaterra, Barcelona, España, **3** ForestWISE—Collaborative Laboratory for Integrated Forest & Fire Management, Vila Real, Portugal, **4** Dirección General de Ecosistemas Forestales y Gestión del Medio (DARP), Barcelona, España, **5** Dipartimento di Scienze Veterinarie, Universitá di Torino, Grugliasco, Torino, Italy

☉ These authors contributed equally to this work.
* taniabarros@ua.pt

**Data Availability Statement:** All data files are available from the GenBank database (ON724047-ON724060.).

## Abstract

The Iberian ibex is one of the most singular species of the Iberian Peninsula. Throughout the years, this species suffered several threats which led the population to its decline. Many reintroductions and translocations were made, however, none of those actions took into account the genetic patterns of both reintroduced individuals and the target populations. In this paper, we explored the genetic traits of three populations of Iberian ibex in Catalonia, which experienced blind reintroductions in past years: The populations of Iberian ibex from *Els Ports de Tortosa i Beseit National Game Reserve* (TBNGR), *Montserrat Natural Park* (Monserrat) and *Montgrí, les Illes Medes i el Baix Ter Natural Park* (Montgrí) Based on the genetic patterns of the three populations coupled with the absence of genetic introgression with domestic goats–inferred using mitochondrial and nuclear markers–we propose that these should be regarded as two different management units: TBNGR coupled with Montserrat, and Montgrí. Montserrat population should be targeted as a population model for ecology and evolution studies. Although we did not detect evidences of recent bottleneck events, this population seems to be monomorphic for the mtDNA haplotype. Our results suggest that the blind reintroductions from TBNGR to Montserrat failed on maximizing the genetic diversity of the latter. We enhance the importance of genetic monitoring of both the source population and the selected individuals to be re-introduced. We conclude that the three studied population of Iberian ibex must be monitored to explore which strategy would be advantageous for maintaining the genetic diversity. On the other hand, TBNGR should be monitored to assess the existence of its singular genetic variation, where stochastic events could preserve this lost genetic variation.

**Funding:** .B. is funded by national funds through CESAM/DBIO under the POSEUR project CP01-MARG-QUERCUS/2018. https://poseur.portugal2020.pt/ E.F. is funded by national funds (OE), through FCT—Fundação para a Ciência e a Tecnologia, I.P., in the scope of the framework contract foreseen in the numbers 4, 5 and 6 of the article 23, of the Decree-Law 57/2016, of August 29, changed by Law 57/2017, of July 19. https://www.fct.pt/ J.F. was supported by a PhD grant from FCT/MCTES (PD/BD/150645/2020), co-financed by European Social Fund POPH-QREN program. https://www.fct.pt/ J.C. was supported by a research contract (CEECIND/01428/2018) from the Fundação para a Ciência e a Tecnologia (FCT). https://www.fct.pt/ FCT/MCTES financed CESAM (UIDP/50017/2020+UIDB/50017/2020 +LA/P/0094/2020), through national funds. https://www.fct.pt/ "Apoio à Contratação de Recursos Humanos Altamente Qualificados" (NORTE-06-3559-FSE-000045), supported by Norte Portugal Regional Operational Programme (NORTE 2020), under the PORTUGAL 2020 Partnership Agreement. ForestWISE—Collaborative Laboratory for Integrated Forest & Fire Management, was recognized as a CoLAB by the Foundation for Science and Technology, I.P. (FCT). https://www.forestwise.pt/pt/institution/ This article/publication is based upon work from COST Action, supported by COST (European Cooperation in Science and Technology). https://www.cost.eu/ The funders had no role in study design, data collection and analysis, decision to publish, or preparation of the manuscript.

**Competing interests:** The authors have declared that no competing interests exist.

## Introduction

The endemic Iberian ibex (*Capra pyrenaica*) is one of the most iconic ungulate species in the Iberian Peninsula [1]. The species was originally distributed throughout the Iberian range, from sea level up to 3,400m, and also in the southwest of France. Currently the species is extinct in the northernmost areas of its native range, namely in the Pyrenees [2]. Four Iberian ibex subspecies have been historically recognized based on coat color and horn morphology [3–5]. Only two of these subspecies are currently extant: *C. p. victoriae* and *C. p. hispanica*, with allopatric distribution in the Iberia Peninsula. The subspecies *C. p. victoriae* occurs in the central Spanish mountains (Sierra de Gredos) and has been reintroduced to a number of additional sites in Spain (Batuecas, La Pedriza, Riaño, French Pyrenees) and naturally re-colonized northern Portugal (Peneda-Gerês National Park) [6, 7]. The subspecies *C. p. hispanica* occupies the arc of mountains that run along the Mediterranean coast, as well as Sierra Morena [8, 9]. The other two subspecies went extinct on different moments: *C. p. lusitanica*, from Iberian NW mountains, in the end of the 19th century; and *C. p. pyrenaica* in the end of 20th century [10, 11]. The latter formerly occurred throughout much of the French, Spanish and Andorran Pyrenees, and persisted until recently in Ordesa and Monte Perdido National Park, in the Maladeta massif [2].

During the 19th century, the Iberian ibex underwent a considerable demographic decline caused by the destruction of its natural habitat, epidemics and unrestrained hunting activities [12]. The populations of Iberian ibex faced numerous threats, including contagious diseases, habitat fragmentation and poaching or unrestricted hunting plans that resulted on biased sex ratio and age structure [13–16]. At the beginning of the 20th century, several conservation programs were at the base of the recovery of the Iberian ibex populations [11, 17, 18]. The species is currently widely distributed, however, the alteration and even loss of genetic diversity due to non-controlled translocations, re-introductions and natural spreading is envisaged as a threat. *C. pyrenaica* is a very popular game species and several translocations and restocking programs were conducted with hunting purposes, although many of these were poorly documented [19, 20]. These unmonitored translocations might have resulted in the admixture of animals from distinct origins. Graver, though, is the possibility of hybridization with domestic goat (*Capra aegagrus hircus*), as recently reported [21, 22], threatening the integrity and fitness of the species gene pool. Additionally, other studies using genotypic data suggest that the Iberian ibex is strongly differentiated among regions, and presents low genetic diversity and poor mixed ancestry, which may be a consequence of past augmentation translocations that resulted in a poor increase of genetic diversity of the species.

Reintroductions have been widely used for several wild species and have been proven to be successful on the restoration of populations that are at risk of decline. It is considered a fruitful conservation tool, although its implementation needs to present consistent justification and it needs to reflect a balance between conservation benefits and its costs [23]. Management practices targeting ungulate species–including translocations, captive-breeding, fencing–potentially lead to changes in the genetic patterns of populations [24–26] but can result on: the decline of genetic variation, increased inbreeding, low population viability and even loss of local adaptations [27–29]. Demographic restoration may not reflect genetic improvement of the target populations [30]. For these reasons, information on the phylogeographic origins and genetic affinities among populations may provide useful insights for the implementation of conservation and management strategies, although this is still underappreciated [31]. Hence, using genetic monitoring as a diagnostic tool, one can acquire valuable information regarding possible founder effects, success of supplementation, population dynamics and admixture, as well as other genetic-related parameters [32–34]. The use of genetic tools do play an important

a very essential role in reintroduction programs and should be implemented prior to the introduction of any specimens into the target population [35].

The populations of Iberian ibex from *Els Ports de Tortosa i Beseit National Game Reserve* (TBNGR), *Montserrat Natural Park* (Monserrat) and *Montgrí, les Illes Medes i el Baix Ter Natural Park* (Montgrí), in Catalonia (Spain), are three populations that have been under active management plans, that involved translocations and population reinforcement measures [36]. The Montserrat and Montgrí populations were originated from single founder events comprising, respectively: (i) 46 individuals translocated from TBNGR to Montserrat, during the 90's [36]; (ii) and five runaway individuals brought to Montgrí from the Sobrestany zoo, in Girona, that were originally from Andalusia. No population genetics studies have ever been envisaged for these three populations in specific, as genetic monitoring of the specimens involved in those translocation measures was not performed. The implementation of molecular methods should be implemented as a tool for monitoring genetic changes in ungulate populations [37], especially in those that are explored for hunting purposes, which is the case of Montserrat, Montgrí and TBNGR populations. Such actions may affect genetic variability in a short period of time [14, 38].

Using both mitochondrial and nuclear markers, we aimed to characterize the genetic patterns of the three mentioned populations and evaluate if the translocation measures improved the genetic variability of the population (in the particular case of Montserrat). Specifically, we (1) assessed the presence and level of hybridization with domestic goat, as well as human-mediated introgression (through translocations or rearing in confined multi-species herds) of Alpine ibex (*Capra ibex*), as it may constitute a threat to the maintenance of genetic heritage of the Iberian ibex; (2) characterized the phylogeographic affinities with other *C. pyrenaica* populations in the Iberian Peninsula; and (3) assessed the level of genetic diversity, structure and relatedness within and between the three Iberian ibex populations. The populations of Montserrat and Montgrí are a result of known historical founder events, thus, the assessment of diversity and endogamy levels are particularly relevant for these two populations. Additionally, given that the population of Montserrat was founded by individuals coming from TBNGR, these two populations may constitute an interesting case study and provide relevant insights into the microevolutionary processes at stake during translocation events. Thus, we also aim to (4) to discuss the genetic patterns observed in TBNGR and Montserrat at the light of the intimate relation among the two populations. These insights may highlight the importance of these and other reintroduced populations for conservation and management actions for this species in the Iberian Peninsula.

## Material and methods

### Study areas

The Iberian ibex populations of TBNGR, Montserrat Natural Park and Montgrí mountain (northeastern Spain, Fig 1), experience a Mediterranean climate characterized by hot summers, summer droughts, and cold winters [39]. The habitat is heterogeneous and encompasses holm oak (*Quercus ilex*) and pine (*Pinus* sp.) forests, evergreen sclerophyll shrublands and patches of pastures and crops. The Iberian ibex census in the TBNGR suggested a population decrease of approximately 25% over the last two decades [40]. Here, Iberian ibex males have been trophy-harvested over the last four decades. Selective harvesting towards individuals with undesired phenotypes is also carried out to reduce the intraspecific competition and to limit their reproduction [15]. The Iberian ibex population of *Montserrat* Natural Park resulted from a reintroduction programme conducted in the 90's, when 46 individuals were translocated from the TBNGR. Currently, a natural increase in population density is taking place and a

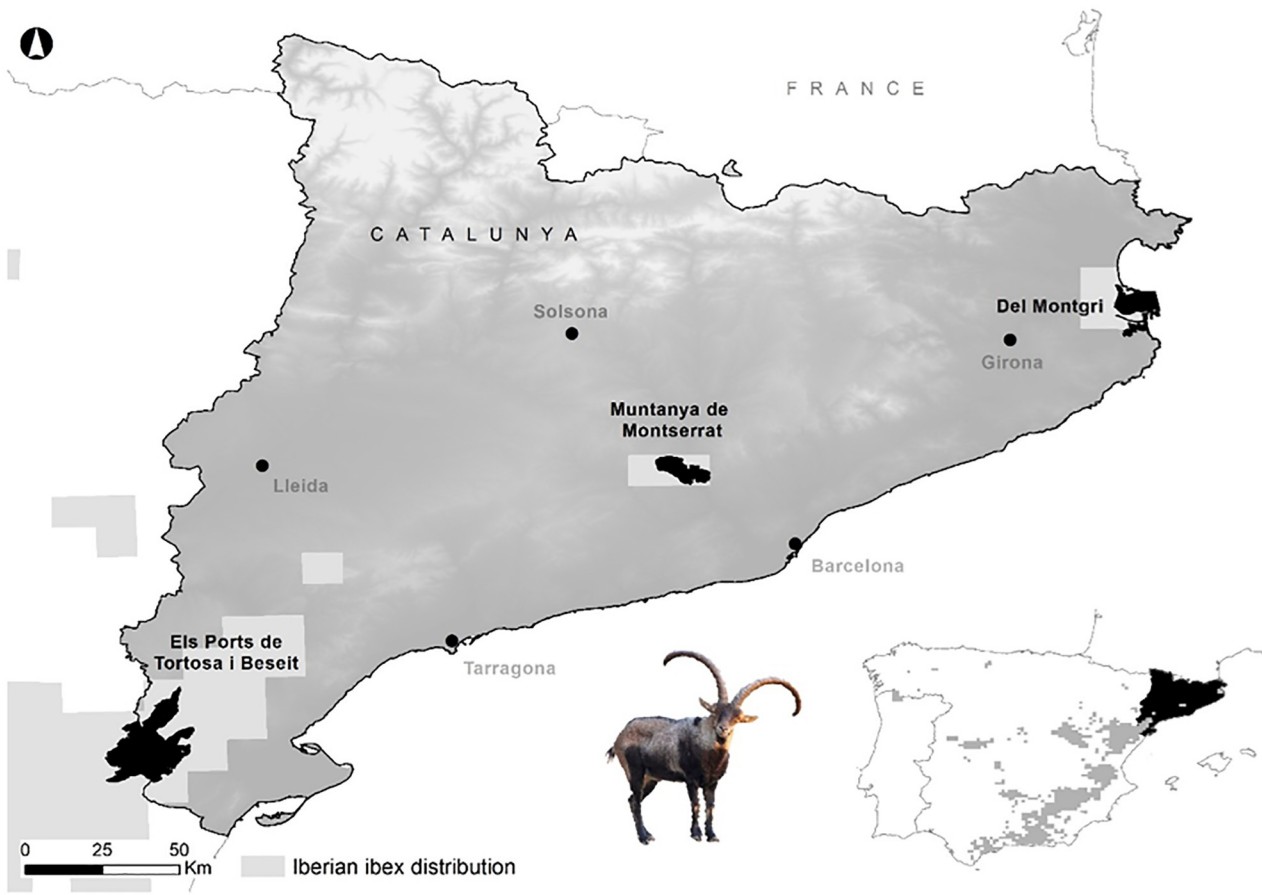

**Fig 1. Location of the three studied populations of Iberian ibex.** Elevation data used in the map were retrieved from the Shutter Radar Topography Mission (SRTM-DEM; [42]), that is publicly available at no charge, a courtesy of the U.S. Geological Survey.

restricted number of males can be hunted once the individual grows enough to be classified as a trophy. Since Iberian ibexes were accidentally released into Montgrí (2000–2007) [41], no further translocations were carried out. Current censuses point to a viable population with ca. 70 individuals [41].

## DNA amplification and sequencing

In this work, Iberian ibex (*Capra pyrenaica*) samples were collected from legally hunted individuals harvested in different populations of the Catalan community. None of the animals were hunted for our scientific purposes and thus an ethic committee approval was not necessary. This sampling was performed within the framework of an official agreement between the Ministry of Climate Action, Food and Rural Agenda (Generalitat de Catalunya, Departament d'Acció Climàtica, Alimentació I Agenda Rural) and the Servei d'Ecopatologia de Fauna Salvatge (SEFaS). This agreement has as a main goal the surveillance of diseases from wildlife from the region of Catalonia, in Spain. None of the authors were directly involved in the sample collection, and the samples were not purposefully collected for this study. Government employers delivered tissue samples to our laboratory. DNA was isolated from a total of 185 muscle samples (118 ibexes from TBNGR, 21 ibexes from Montserrat, 8 ibexes from Montgri, 31 domestic goats and 7 ibexes without known geographic coordinates) through a standard

salt-extraction protocol [43]–or using the QIAGEN® Tissue and Blood extraction kit, for older samples–and quantified by spectrophotometry, using Nanodrop®. We amplified a fragment of 1140 bp from mitochondrial DNA (mtDNA; cytochrome b) using the universal primers L14841 and H15915 [44]. We amplified the target region, in a final volume of 25 microliters with 1x reaction buffer, 1.5 mM $MgCl_2$, 1 μM of each primer, 10 mM of dNTP, 0.2 units of Taq polymerase (Invitrogen©). We performed a PCR amplification with 45 cycles (denaturation: 94˚C, 3 minutes; annealing: 48˚C, 45 seconds; polymerization: 72˚C, 120 seconds). Amplified (PCR) products were visualized in 2% agarose gels, purified and sequenced (Sanger) in an automatic sequencer ABIPRISM® 3730-XL DNA Analyzer, Applied Biosystems™, using BigDye™ sequencing kit.

A total of 14 microsatellite markers were selected for this study (Supporting Material S1 File). DNA amplifications were performed using the QIAGEN® Multiplex amplification kit, following manufacturer's conditions. PCR products were visualized on 2% agarose gel and fragment analysis was performed using an ABIPRISM® 3730-XL DNA Analyser from Applied Biosystems™. In order to reduce the chance of mistype and minimize genotyping error, each sample was amplified and genotyped a minimum of three times for each marker.

## Data analysis

Generated sequences were aligned and consensus sequences were inferred from the forward and reverse sequences. Sequences were then compared with previously published sequences of Iberian ibex, Alpine ibex, Nubian ibex (*C. nubiana*), Siberian ibex (*C. sibirica*) and domestic goat, retrieved from Genbank (Supporting Material S2 File). Mitochondrial DNA sequences were aligned using MEGA v.5 [45] with the CLUSTALW algorithm [46] and were further manually edited. Preliminary species diagnosis (for a preliminary screen of hybridisation with *Capra a. hircus*) was performed using BLAST queries to the GenBank® online database. A haplotype network was constructed using a median-joining (MJ) algorithm implemented in POPART [47]. General diversity indices were calculated on DNAsp v5, using the generated mtDNA sequences: number of haplotypes, haplotype and nucleotide diversity and number of polymorphic sites [48].

Microsatellite genotyping was performed using Genemarker™ v2.4.1 (SoftGenetics™). Electropherograms were analysed with this software but allele calling was performed through careful visual inspection. Identification of individual profiles was assessed only when at least 70% of the analysed microsatellites were successfully amplified. Mean genotyping error rate was estimated by assessing the ratio between the number of mistyped alleles relatively to the total number of genotyped alleles [49].

For detecting structure and possible hybridization between the Iberian ibex and domestic goat, we tested for evidence of genetic structure using Structure v2.3.4 [50]. We used the admixture model with correlated allele frequencies with no previous information about the original sampled location of each individual. The analysis was carried out for 2 million iterations of the Markov Chain Monte Carlo, with a burn-in of 100 thousand iterations. The putative number of populations was simulated with K (number of putative subpopulations) varying between 1 and 6. Ten replicate runs were performed for each K. We then used Structure Harvester [51] to summarize the results, by estimating the best K using both the Evanno method [52] and the original approach suggested by Pritchard et al. [50]. We then performed a Structure analysis within the Iberian ibex population, following the same conditions as described above. Evidence for significant deviations from Hardy-Weinberg equilibrium (HWE), and for linkage disequilibrium (LD), as well as allelic frequencies and patterns, heterozygosity levels and genetic differentiation through $F_{ST}$ index [53] were assessed. Analysis of

molecular variance (AMOVA) using the allelic distances between individuals was also calculated. All analyses were performed on Genalex v6.5 [54] and Arlequin v3.5 [55].

In order to search for evidence of recent bottlenecks, we estimated the $M$ ratio between the number of alleles and the allele size range [56]. We estimated this ratio using ARLEQUIN, after we recoded the genotypes using the number of tandem repeats relatively to the smallest allele of each locus (which was coded with 1, for one repeat).

## Results

### Mitochondrial DNA data analysis

We were able to successfully sequence a total of 164 amplified samples (102 samples from TBNGR, 21 samples from Montserrat, 8 samples from Montgrí, 28 samples from domestic goats and 5 samples without geographic coordinates) for mtDNA, comprising 14 Iberian ibex haplotypes (Fig 3; Table 1; Accession numbers: ON724047-ON724060). All but two haplotypes (H10 and H12) were newly generated in this study and had not been detected in previous studies. Haplotypes plotting across the sampling area evidences a wide distribution of haplotype H10, corresponding to the most frequent haplotype found in the studied area (present in ca. 64% of all analyzed samples). This haplotype is present in the population located in the TBNGR and it was the only haplotype detected in individuals sampled in Montserrat. Haplotype H8 is present in 10% of the total samples and was reported only for individuals from TBNGR. Haplotype H12, also present in 10% of the total samples, was only detected in individuals sampled in Montgrí (Fig 2).

With the aim of balancing the number of retrieved sequences and the amount of information per sequence (i.e., covering the maximum number of polymorphic sites found in previously published sequences from the selected species available on Genbank), the haplotype network was built using 81 complete sequences with a minimum length of ca. 620 base pairs. All haplotypes generated in this study clustered with Iberian ibex individuals rather with domestic goat or Alpine ibex, as expected in the case of absence of hybridization between the wild and domestic populations. Nevertheless, we must call attention to the fact that mtDNA refers to matrilineal non-recombinant lineages and thus, evidence against hybridization, in

**Table 1. Correspondence between presence of the found haplotypes in the studied population of Iberian ibex, by sampling area and by genetic cluster (according to the Structure results for k = 2).**

| Haplotypes | By sampling area | | | By genetic cluster | |
| --- | --- | --- | --- | --- | --- |
| | Tortosa i Beseit | Montserrat | Montgrí | Tortosa i Beseit + Montserrat | Montgrí |
| H1 | 1 | | | 1 | |
| H2 | 1 | | | 1 | |
| H3 | 1 | | | 1 | |
| H4 | 1 | | | 1 | |
| H5 | 1 | | | 1 | |
| H6 | 1 | | | 1 | |
| H7 | 1 | | | 1 | |
| H8 | 7 | | | 7 | |
| H9 | 1 | | | 1 | |
| H10 | 30 | 15 | | 45 | |
| H11 | 1 | | | 1 | |
| H12 | | | 7 | | 7 |
| H13 | | | 1 | | 1 |
| H14 | | | 1 | | 1 |

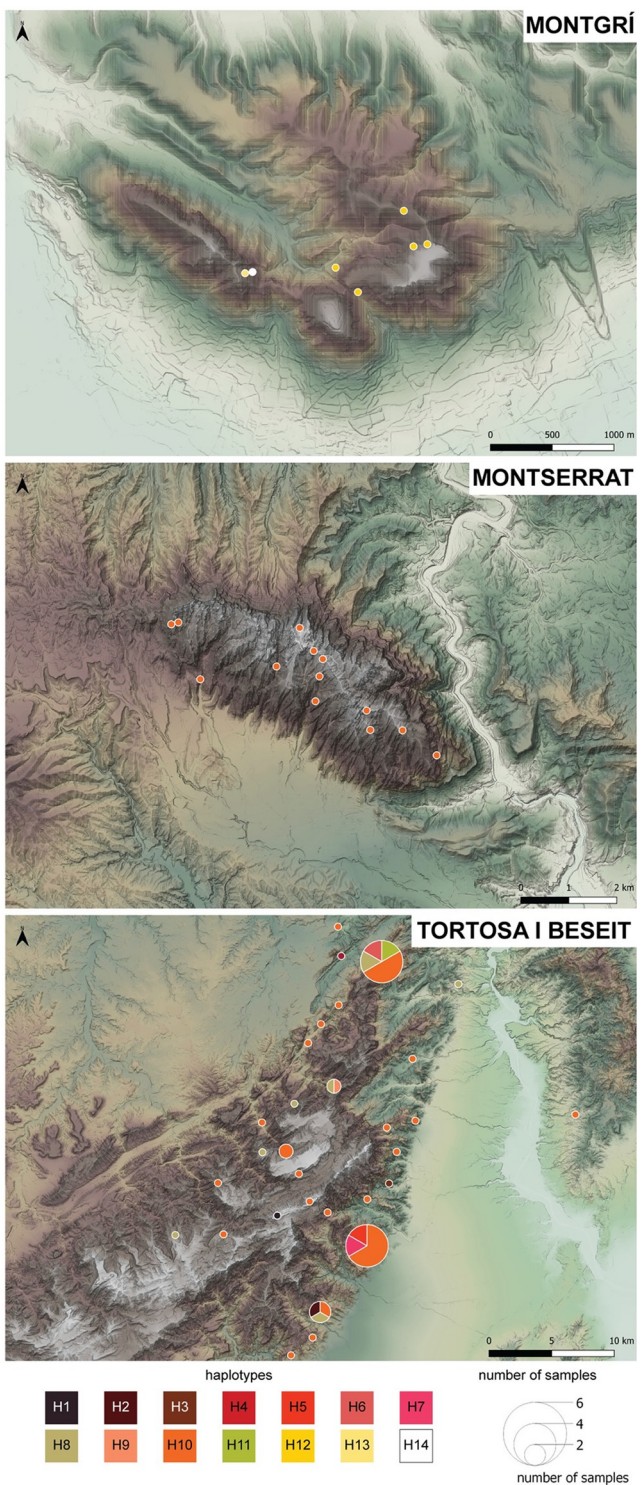

**Fig 2. Haplotype distribution of the Iberian ibex samples across the studied area.** Each haplotype (H1 to H14) is identified by a different color. Figure adapted from MDT05 2015 CC-BY 4.0 scne.es.

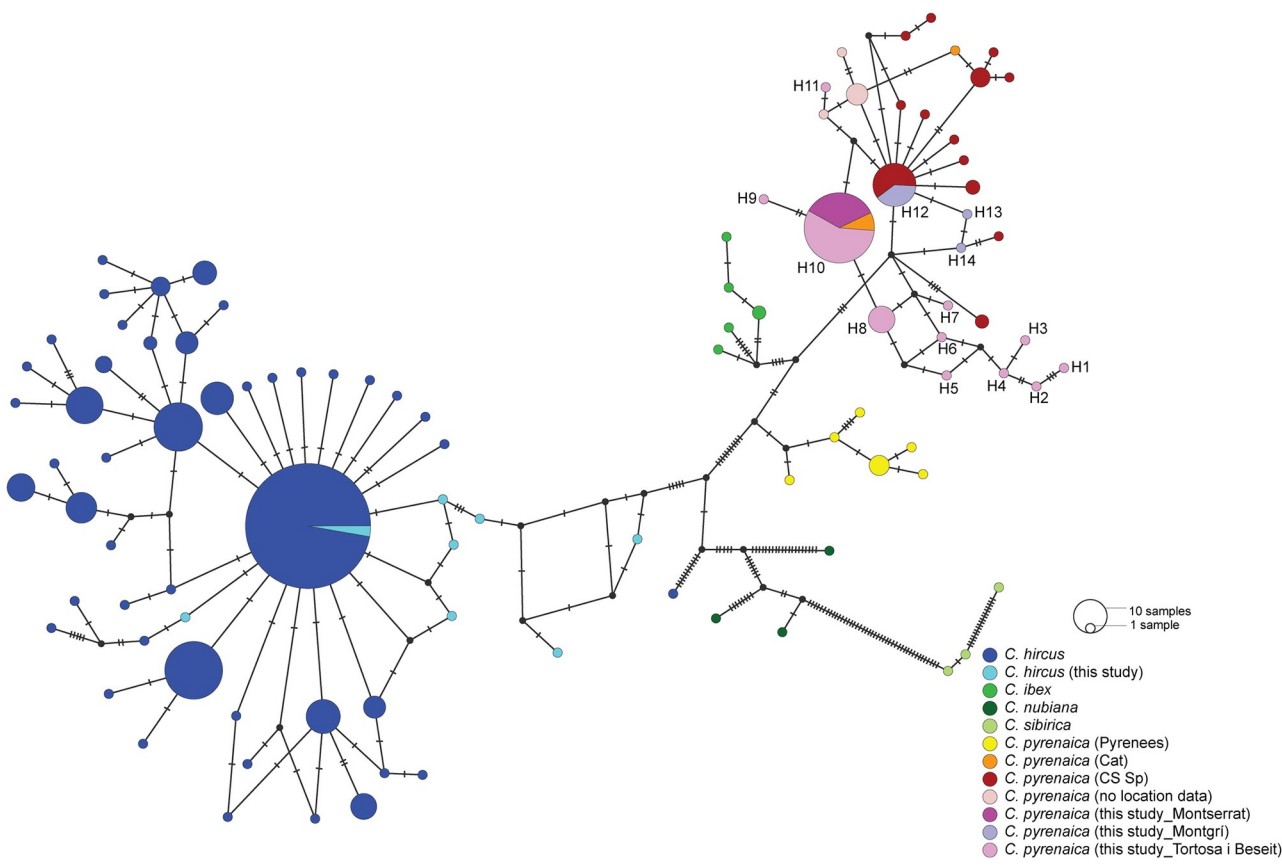

**Fig 3. Haplotype network showing the haplotypes of *Capra pyrenaica* and *Capra a. hircus* identified in this study, and previously published haplotypes of Iberian ibex, Alpine ibex and domestic goat.** (Cat–Catalonia; CS_Sp–Centre/Southern Spain).

this case, is limited. Contrarily to what we would expect, the inferred phylogeographic affinities within *C. pyrenaica* and with closely related species, did not support the monophyly of the species. In fact, *Capra ibex* appears as the sister group to all the *C. pyrenaica* subspecies, except the type subspecies, *C. p. pyrenaica*. We found shared haplotypes among TBNGR, Montserrat and previously published sequences of individuals from the same geographic range (Catalonia). Goats from Montgrí, were found to share haplotypes with individuals from Central and Southern Spain (Fig 3).

## Microsatellite data analysis

We found a clear structure between *C. a. hircus* and *C. ibex* sampled in the study area (Fig 4A), with percentages of attribution to each one of the clusters above 99% for each individual (Fig 4D) (Supporting Material S3 File). Regarding the Iberian ibex genotypes, we were able to retrieve a total of 96 individual genotypes (Supporting Material S4 File). We also found structure among the Iberian ibex populations under study, in particular between the population from Montgrí and the populations of TBNGR and Montserrat (Fig 4E). According to the Evanno method, the most likely scenario within the Iberian ibex population from Catalonia comprises two genetic clusters (K = 2), with a *Q* proportion of the genome of each individual assigned to each of the inferred clusters above 96%. One cluster gathers all individuals from TBNGR and Montserrat, while all Montgrí individuals (Fig 4B) are grouped within the other genetic cluster. On the other hand, when inferring the most likely *K* number of genetic clusters

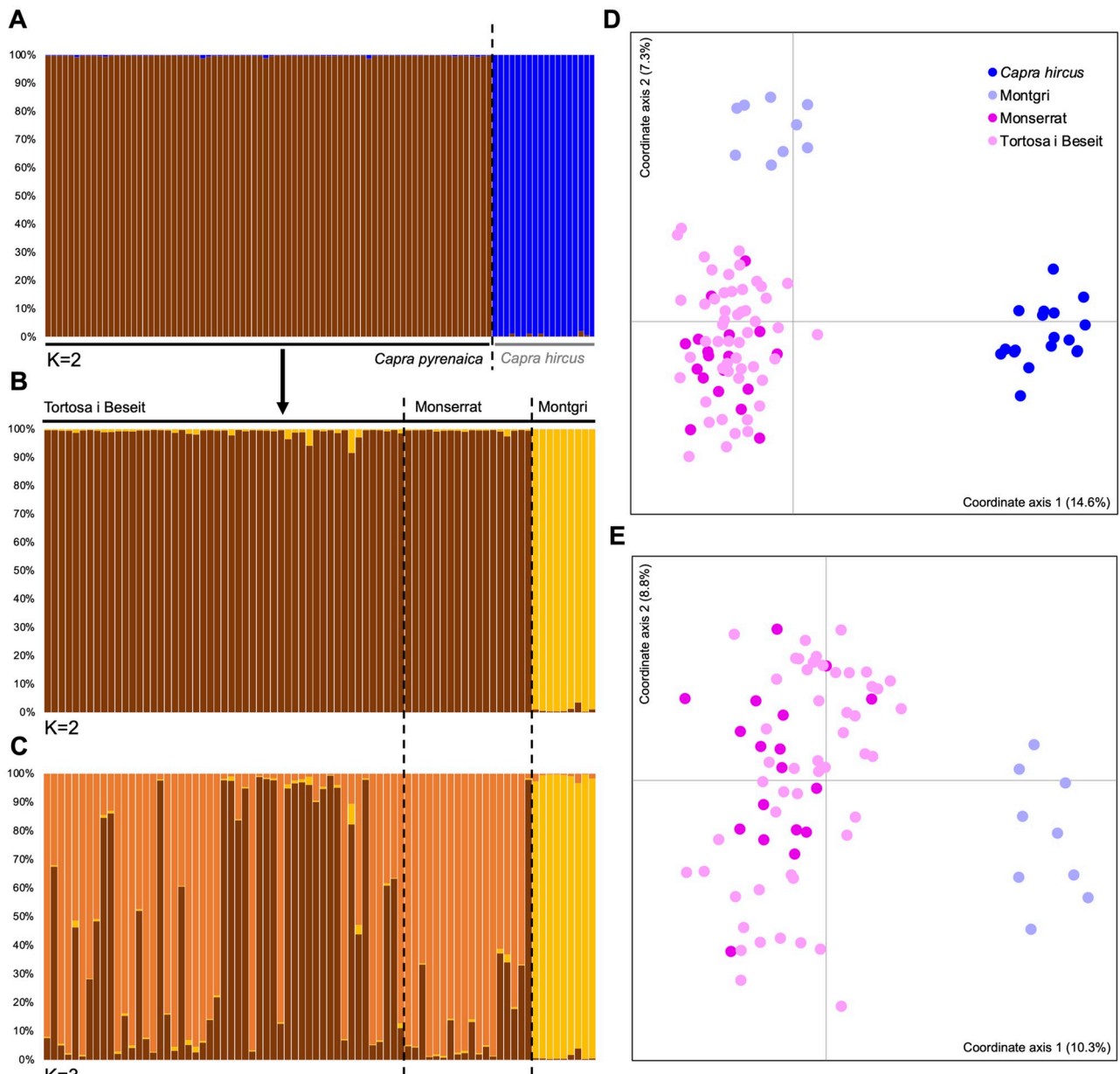

**Fig 4. Microsatellite analysis for both domestic goats (*C. a. hircus*) and the three Iberian ibex populations sampled in Catalonia.** Individual *Q* proportion of the genome of each individual assigned to each of the genetic clusters inferred with STRUCTURE for (A) *C. a. hircus* and *C. ibex* sampled in the study area, and for (B) K = 2 and (C) K = 3, for the three Iberian ibex populations sampled in Catalonia. Each individual is represented by a vertical bar and the different genetic clusters are represented by different colors. Principal coordinates analysis (PCoA) plot for (D) *C. a. hircus* and *C. ibex* sampled in the study area and for the three Iberian ibex populations samples in Catalonia.

directly from likelihood values (following the original approach from Pritchard et al. 2000), the scenario with higher support includes with three genetic clusters (K = 3) of Iberian ibex, as follows: (i) one comprising all individuals sampled in Montgrí; (ii) a second one comprising almost all individuals sampled in Montserrat and part of the individuals from TBNGR; and (iii) a third composed only by individuals from TBNGR, with exception to an individual from Monserrat (Fig 4C).

**Table 2. Genetic diversity, structure, levels of endogamy (F$_{IS}$) and bottleneck results for the studied populations of Iberian ibex.**

| | | Parameter | *Catalonia (n = 78)* | *Tortosa i Beseit (n = 51)* | *Montserrat (n = 18)* | *Montgrí (n = 9)* |
|---|---|---|---|---|---|---|
| **Mitochondrial DNA** | **Genetic diversity** | **π** | 0.003 | 0.004 | - | 0.001 |
| | | **Hd** | 0.573 | 0.604 | - | 0.524 |
| | | **S** | 14 | 14 | - | 2 |
| **Microsatellites** | **Structure** | **Loci in HWD** | 8/14 | 2/14 | 1/14 | 1/14 |
| | | **LD (pair of loci in LD)** | 11/91 | 3/91 | 0/91 | 1/91 |
| | **Genetic Diversity** | **NA** | 7 | 5 | 4 | 3 |
| | | **A$_P$** | - | 24 | 9 | 8 |
| | | **A$_r$** | 3.073 | 5.286 | 4.143 | 2.714 |
| | | **Gene Diversity** | 0.652 | 0.641 | 0.560 | 0.421 |
| | | **H$_E$** | 0.659 | 0.574 | 0.670 | 0.351 |
| | | **H$_O$** | 0.569 | 0.627 | 0.629 | 0.485 |
| | **Endogamy** | **F$_{IS}$** | **0.155** | **0.114** | -0.157 | **0.219**$^*$ |

Bold values indicate significance (p<0.001) or equivalent, after Bonferroni correction, applied for multiple testing.

$^*$—marginally significant.

π–nucleotide diversity; Hd–haplotype diversity; S–number of polymorphic sites; HWD–Hardy Weinberg Disequilibrium; LD–Linkage Disequilibrium; NA–number of alleles (average among loci), Ap–number of private alleles; Ar–allelic richness (average among loci); HE–expected heterozygosity (average among loci); HO–observed heterozygosity (average among loci). Montserrat population was monomorphic for mtDNA.

F$_{ST}$ values show a very strong and significant genetic differentiation between the population pairs TBNGR/Montgrí (F$_{ST}$ = 0.25, p<0.001) and Montserrat/Montgrí (F$_{ST}$ = 0.29, p<0.001). There was no evidence of a significant genetic differentiation between TBNGR and Montserrat (F$_{ST}$ = 0.01, p<0.001). AMOVA results showed that for the three populations ca. 80% of the genetic variation is found within the individuals, ca. 5.5% among individuals and ca. 14% among populations.

Number of polymorphic sites, nucleotide and haplotype diversity were calculated for the overall population of Iberian ibex and also for each population separately (Table 2). We have sampled 11 out of the 14 identified haplotypes in TBNGR, while in Montserrat we only identified one haplotype. Three haplotypes, not shared with the other two populations, were sampled in Montgrí (Fig 3). Haplotype and nucleotide diversity was found to be higher in TBNGR. However, such results must be interpreted with caution, as both parameters are sensitive to the number of samples. As expected, microsatellite genotypes revealed a high level of departure from HWE for the whole population of Iberian ibex in Catalonia, as well as strong evidence of linkage disequilibrium (LD). When considering the three subpopulations individually, TBNGR carries the higher number of loci departing from HWE, as well as LD. Number of private alleles, allelic richness and gene diversity were also found to be higher in this population. We also found evidence of inbreeding in TBNGR and in Montgrí, but only marginally significant in the latter (Table 2). The significant departures of HWE and presence of LD in the three populations might be related with natural variation (TBNGR and Montserrat are, in fact, one single genetic cluster, though not the same reproductive unit) and the reduced number of samples, in this latter case the Montgrí population, and other stochastic causes.

Since two of the microsatellite markers (BM1258, IDVGA30) had substantially more missing data than the other twelve markers, and that this might result in the estimation of artificially low values of *M* ratio, this ratio was estimated using only twelve markers. The values ranged from 0.623, in Monserrat, to 0.682, in Montgri. These values are very close to the theoretical threshold of *M* = 0.68, inferred by the authors of the method [56] as providing evidence of recent population declines. These results are consistent with the known demographic

history of these populations, given that at least Montgri and Montserrat originated from known founder events. More details on microsatellite markers diversity indices are provided in supporting material (S5 File).

## Discussion

Recent studies suggested that genetic monitoring is a key tool to be applied in reintroduction and translocation actions [57]. Unfortunately, the majority of monitoring programs do not use the full potential given by molecular genetic markers, which can provide information relevant to both ecological and evolutionary traits [58]. The case of these three Iberian ibex populations constitutes a valuable opportunity not only to assess the current genetic patterns and detect possible admixture and genetic structure, but also if past translocation actions were effective for maximizing genetic diversity.

### Evolutionary affinities and current gene flow with other Capra species

Our results suggest that there is no introgression from domestic goat or Alpine ibex gene pools in the three surveyed Iberian ibex populations. The patterns of assignment to genetic clusters inferred from the recombinant nuclear markers (microsatellites) corroborate the results of the phylogenetic analysis based on mtDNA and clearly differentiated Iberian ibex from domestic goat. We identified the presence of two clearly distinct genetic clusters separating Iberian ibex individuals from domestic goat individuals with a $Q$ proportion of the genome of each individual assigned to each genetic cluster, matching its putative source population by more than 99% (see Supporting Material S3 File). Both mitochondrial and microsatellite results indicate that there is no evidence of hybridization between both species in the populations under study (Montgrí, Montserrat and TBNGR), a scenario that was already detected in other areas of the distributional range of this species [58].

On the other hand, the inferred evolutionary relationships reveal the non-monophyletic nature of *C. pyrenaica*, in relation to *C. ibex*. The possible presence of *C. pyrenaica* in the Iberian Peninsula and its differentiation in the Late Pleistocene, or the fact that introgression by hybridization is a common process in the evolution of the *Capra* genus [59], may justify the paraphyletic or polyphyletic nature, respectively, of the Iberian ibex. In fact, either the historical hybridization with *C. ibex* and other ancient *Capra* taxa [60], or the distinct times for the colonization of the Iberian by the ancestors of *C. p. pyrenaica* and the other Iberian ibex subspecies may help to understand these phylogenetic patterns. Also, skull morphology revealed an intermediate position of *C. p. pyrenaica* between the other 2 extant subspecies (*C. p. victoriae* and *C. p. hispanica*) and the Alpine ibex (*C. ibex*) [60]. Genetic studies pointed that the Pyrenean wild goat is clearly differentiated from the other Iberian populations [17]. Under the light of the evolutionary relations among Iberian ibex and Alpine ibex [61], our results leave room for questioning the taxonomic relations of European *Capra* species. In addition to providing new data from Iberian goats geographically closer to the ancient range of *C. p. pyrenaica*, mtDNA phylogeography clearly supports the polyphyletic nature of what has been referred to as *C. pyrenaica*, by placing the Alpine ibex in-between the Iberian ibex and the Pyrenean "bucardo".

### Genetic diversity patterns and structure of the Iberian ibex in Catalonia

Genetic structure patterns inferred with nuclear DNA, are in agreement with mtDNA observed patterns, not only at the level of separating domestic from wild goat, but also between the wild goat populations. The high nuclear genetic diversity found in TBNGR was expected, since this population did not suffer from an intense genetic bottleneck as the other populations

that were originated by a low number of individuals (46 individuals in Montserrat and 5 individuals in Montgrí). On the other hand, mitochondrial genetic diversity was found to be higher in Montgrí, yet, this result should be interpreted with caution, as several diversity indices are sensitive to sample size.

The wild goat population from Montgrí was originated by a group of Iberian ibex individuals translocated from Andalusia, that escaped from a zoo in Sobrestany, Girona province [41]. Hence, the presence of exclusive haplotypes, the sharing of certain haplotypes with goats from Centre and Southern Spain, the presence of private alleles and distinct allele patterns, that differ from those observed in TBNGR and Montserrat. Further, the strong geographic isolation of Montgrí population and the absence of natural or artificial translocation of individuals from the other two populations might be related with the higher number of private alleles and the high level of endogamy found in this population. However, and most importantly, the strong founder effect that originated the Montgrí population might be the pivotal cause underlying its observed allelic patterns, which may raise questions regarding the management of this population.

The observed structure between the Iberian ibex populations from Montgrí and from TBNGR and Montserrat are supported by the estimated high $F_{ST}$ values. High $F_{ST}$ values were already observed in other studies of within the Iberian ibex subspecies [58]. The genetic affinities between the source (TBNGR) and newly founded (Montserrat) populations is supported by either of the two most likely scenarios: K = 2 or K = 3 (Fig 4).

Regarding K = 3 results, the individuals from both populations were not spatially segregated between Montserrat and TBNGR. Rather, there was high values of $Q$ proportion of the genome of each individual, from both populations, assigned to both clusters. One of the two clusters, however, was more associated with the population from Montserrat. These results show the strong founder effect in Montserrat that resulted in a new random sample of alleles and allele frequencies and that, after 25 years since the reintroduction of the first individuals, Montserrat population might be in its way into becoming a new random mating population, genetically distinct from the population it was originated from. Thus, as previously reported [36], the population of wild goats from Montserrat, in comparison with the source population of TBNGR, is likely to become a good model for studies in ecology and evolution. Reproductive dynamics may also reflect on the genetic composition, hence, the structuration of the populations after reintroduction can give important clues about those reproductive dynamics [61]. A detailed monitoring of this reintroduced population from Montserrat could provide the necessary baseline information to examine the effects of human-induced and/or environmental changes on phenotypic traits variation and quantify the effects of variable nutrition and dietary restrictions on secondary sexual characters, such as horn growth and other traits.

It is important to notice that, despite the absence of genetic differentiation between TBNGR and Montserrat–that derives from the fact that Montserrat population was originated by individuals from TBNGR–the evidences of independent reproductive dynamics of the two populations start to leave a genetic signature on both populations' gene pools. However, due to the number of founder individuals in Montserrat and the matrilineal lineages in TBNGR, it was not expected that the Montserrat population would be monomorphic for mtDNA. Despite the absence of inbreeding and bottleneck in Montserrat, both molecular markers tell distinct stories: while mtDNA reveals a low genetic diversity (only one haplotype), microsatellites reveal an allelic richness and heterozygosity values similar with those from original population (TBNGR). This raises questions regarding if this outcome is an effect of the selection of specific features associated with particular genetic lineages (particularly matrilineal ones), since the founder event, or if it is the result of random genetic drift processes during and after the reintroduction event. This possible current differentiation process between the populations of

Tortosa i Beseit and Montserrat is in part evidenced by the STRUCTURE results and, eventually, by the monomorphic aspect of the mtDNA diversity of Montserrat population.

Other studies on extant populations of *C. pyrenaica* subspecies showed that the Iberian ibex has suffered recent genetic bottlenecks in some locations, which could ultimately and negatively affect the future conservation of this taxon [e.g. 59, 62]. A recent genetic study of the Iberian ibex in southern Spain suggested that severe bottlenecks may have occurred, based on the low mtDNA haplotype and nucleotide diversity estimations [63]. In our study, the values of the *M* ratio, between the number of alleles and the allele size range estimated for the three populations are close to the threshold for populations that gone through recent population bottlenecks [56]. In fact, these values are similar or lower to those estimated for other populations of ungulates or carnivores known to have gone through recent population reductions [64–66].

## Conservation and management implications

The authors of previous studies argued that there are independent local populations of the Iberian ibex in Spain that should be considered as harbours of unique portions of the total genetic variation of the species and they recommended that they should be managed separately, as single management units [15, 67]. Based on the genetic patterns detected in each of the three populations of Iberian ibex, together with the absence of genetic introgression with domestic goats (which could raise concerns), we suggest that these populations should be viewed as different management units: (a) TBNGR together with Montserrat, and (b) Montgrí. However, Montserrat should be regarded as a population model for ecology and evolution studies. Despite the bottleneck events (through the studied nuclear markers), the population seems to be monomorphic for one of the mtDNA haplotypes. These results stress the importance of genetic monitoring of both the source population and the selected individuals to be re-introduced, as the translocation actions from TBNGR to Montserrat failed on maximizing the genetic diversity of the latter. We also point out the case of Montgrí, which is known to have a distinct and "allochthonous" origin, which reflects on the haplotypes and alleles present in the population.

Regarding TBNGR and Montserrat populations, due to their identical origin, we found that both populations may be managed equally, however, special conservation measures should be taken in account for Montserrat population due to their poor genetic diversity concerning mtDNA. We believe that this population will gradually differ from Tortosa i Beseit, even if occasional natural translocations might occur. In fact, translocations may be necessary and can decrease inbreeding and increase genetic variation [31], but those should be performed under a careful inspection of the genetic diversity of the translocated individuals to improve the genetic structure of the population. Under the light of the usage of genetic tools for wild species management, we suggest that in the future these populations may be seen as Management Units [64], as they may play an important role for management and conservation of the Iberian ibex in the Iberian Peninsula, aiding on short-term management of the more inclusive Evolutionary Significant Units [64]. In a nutshell, we believe the three studied population of Iberian ibex should be continuously monitored to test which strategy would be beneficial for the maintenance of genetic diversity (e.g. release of different breeding lines at the same time versus ongoing supplementation) [57], to assess the magnitude of admixture when translocations occur, and assess genetic diversity and population size through time. Finally, we must look for the existence of a part of the lost or "forgotten" genetic variation of the goats from TBNGR, in the neighbouring territories of northern Castellón and Eastern Teruel, where stochasticity could have preserved this lost genetic variation.

## Supporting information

**S1 File. Microsatellite markers used in the study, with reference, Multiplex, GenBank Accession number (GB acc.numb.) and Annealing Temperature (Ta).**
(DOCX)

**S2 File. GeneBank accession numbers (GB acc. numb.) for the *cyt b* sequences from domestic goat (*C. a. hircus*), Alpine ibex (*C. ibex*) and Iberian wild goat (*C. pyrenaica*) used in both the haplotype network and phylogenetic tree.**
(DOCX)

**S3 File. *Q* proportion of the genome of Iberian ibexes (*C. pyrenaica*) and domestic goats (*C. a. hircus*) to each of the two genetic clusters (Cluster 1 and Cluster 2) inferred using STRUCTURE, in the assessment of hybridization among wild and domestic goats.**
(DOCX)

**S4 File. Multilocus genotypes of *Capra hircus* and *Capra pyrenaica* (from individuals sampled at Montgrí, Montserrat and Tortosa I Beseit areas) generated in this study.**
(XLSX)

**S5 File. Number of alleles and level of heterozygosity for the Iberian ibex from Catalonia assessed in this study.** (n = number of individuals; $N_A$ = number of alleles; Ho = Observed heterozygosity; He = Expected heterozygosity; *—monomorphic locus for that population).
(DOCX)

## Acknowledgments

We are grateful Josep Vicens Jovani, Xavier Oliver, Maria Josep Vargas, Xavier Sempere, Jordi Xifra and Ignasi de Dalmases, from the Tortosa and Beseit National Game Reserve, the Controlled Game Area of Montserrat, the Game and Fish Section of Girona (DARP) and the Rural Agents corp, for their advice and samples provided.

## Author Contributions

**Conceptualization:** Tânia Barros, Joana M. Fernandes, Eduardo Ferreira, João Carvalho, Carlos Fonseca, Jordi Ruiz-Olmo, Emmanuel Serrano.

**Data curation:** João Carvalho, Marta Valldeperes, Santiago Lavín, Jordi Ruiz-Olmo, Emmanuel Serrano.

**Formal analysis:** Tânia Barros, Joana M. Fernandes, Eduardo Ferreira.

**Investigation:** Tânia Barros, Joana M. Fernandes, Eduardo Ferreira, João Carvalho, Santiago Lavín, Emmanuel Serrano.

**Methodology:** Tânia Barros, Joana M. Fernandes, Eduardo Ferreira, João Carvalho, Marta Valldeperes, Jordi Ruiz-Olmo, Emmanuel Serrano.

**Project administration:** Emmanuel Serrano.

**Supervision:** Emmanuel Serrano.

**Validation:** Tânia Barros, Eduardo Ferreira, João Carvalho, Marta Valldeperes, Santiago Lavín, Carlos Fonseca, Jordi Ruiz-Olmo, Emmanuel Serrano.

**Visualization:** Marta Valldeperes.

**Writing – original draft:** Tânia Barros, Joana M. Fernandes, Eduardo Ferreira, João Carvalho, Marta Valldeperes, Santiago Lavín, Carlos Fonseca, Jordi Ruiz-Olmo, Emmanuel Serrano.

**Writing – review & editing:** Tânia Barros, Joana M. Fernandes, Eduardo Ferreira, João Carvalho, Marta Valldeperes, Santiago Lavín, Carlos Fonseca, Jordi Ruiz-Olmo, Emmanuel Serrano.

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
