## [Decision Letter · Decision Letter 0]

31 Dec 2021

PONE-D-21-37151Genetic signature of blind reintroductions of Iberian ibex (Capra pyrenaica) in Catalonia, Northeast SpainPLOS ONE

Dear Dr. Barros,

Thank you for submitting your manuscript to PLOS ONE. After careful consideration, we feel that it has merit but does not fully meet PLOS ONE’s publication criteria as it currently stands. Therefore, we invite you to submit a revised version of the manuscript that addresses the points raised during the review process.

We look forward to receiving your revised manuscript.

Kind regards,

Tzen-Yuh Chiang

Academic Editor

PLOS ONE

Journal Requirements:

(T.B. is funded by national funds through CESAM/DBIO under the POSEUR project CP01-MARG-QUERCUS/2018. https://poseur.portugal2020.pt/

E.F. is funded by national funds (OE), through FCT—Fundação para a Ciência e a Tecnologia, I.P., in the scope of the framework contract foreseen in the numbers 4, 5 and 6 of the article 23, of the Decree-Law 57/2016, of August 29, changed by Law 57/2017, of July 19. https://www.fct.pt/

J.F. was supported by a PhD grant from FCT/MCTES (PD/BD/150645/2020), co-financed by European Social Fund POPH-QREN program. https://www.fct.pt/

J.C. was supported by a research contract (CEECIND/01428/2018) from the Fundação para a Ciência e a Tecnologia (FCT). https://www.fct.pt/

FCT/MCTES financed CESAM (UIDP/50017/2020), through national funds. https://www.fct.pt/

“Apoio à Contratação de Recursos Humanos Altamente Qualificados” (NORTE-06-3559-FSE-000045), supported by Norte Portugal Regional Operational Programme (NORTE 2020), under the PORTUGAL 2020 Partnership Agreement. ForestWISE—Collaborative Laboratory for Integrated Forest & Fire Management, was recognized as a CoLAB by the Foundation for Science and Technology, I.P. (FCT). https://www.forestwise.pt/pt/institution/

This article/publication is based upon work from COST Action, supported by COST (European Cooperation in Science and Technology). https://www.cost.eu/

The funders had no role in study design, data collection and analysis, decision to publish, or preparation of the manuscript.)

(T.B. is funded by national funds through CESAM/DBIO under the POSEUR project CP01-MARG-QUERCUS/2018. E.F. is funded by national funds (OE), through FCT—Fundação para a Ciência e a Tecnologia, I.P., in the scope of the framework contract foreseen in the numbers 4, 5 and 6 of the article 23, of the Decree-Law 57/2016, of August 29, changed by Law 57/2017, of July 19. J.F. was supported by a PhD grant from FCT/MCTES (PD/BD/150645/2020), co-financed by European Social Fund POPH-QREN program. J.C. was supported by a research contract (CEECIND/01428/2018) from the Fundação para a Ciência e a Tecnologia (FCT). Thanks are due to FCT/MCTES for the financial support to CESAM (UIDP/50017/2020), through national funds. This work is also a result of the project “Apoio à Contratação de Recursos Humanos Altamente Qualificados” (NORTE-06-3559-FSE-000045), supported by Norte Portugal Regional Operational Programme (NORTE 2020), under the PORTUGAL 2020 Partnership Agreement. ForestWISE—Collaborative Laboratory for Integrated Forest & Fire Management, was recognized as a CoLAB by the Foundation for Science and Technology, I.P. (FCT). This article/publication is based upon work from COST Action, supported by COST (European Cooperation in Science and Technology). We grateful Josep Vicens Jovani, Xavier Oliver, Maria Josep Vargas, Xavier Sempere, Jordi Xifra and Ignasi de Dalmases, from the Tortosa and Beseit National Game Reserve, the Controlled Game Area of Montserrat, the Game and Fish Section of Girona (DARP) and the Agents Rurals body, for their advice and samples provided.)

(T.B. is funded by national funds through CESAM/DBIO under the POSEUR project CP01-MARG-QUERCUS/2018. https://poseur.portugal2020.pt/

E.F. is funded by national funds (OE), through FCT—Fundação para a Ciência e a Tecnologia, I.P., in the scope of the framework contract foreseen in the numbers 4, 5 and 6 of the article 23, of the Decree-Law 57/2016, of August 29, changed by Law 57/2017, of July 19. https://www.fct.pt/

J.F. was supported by a PhD grant from FCT/MCTES (PD/BD/150645/2020), co-financed by European Social Fund POPH-QREN program. https://www.fct.pt/

J.C. was supported by a research contract (CEECIND/01428/2018) from the Fundação para a Ciência e a Tecnologia (FCT). https://www.fct.pt/

FCT/MCTES financed CESAM (UIDP/50017/2020), through national funds. https://www.fct.pt/

“Apoio à Contratação de Recursos Humanos Altamente Qualificados” (NORTE-06-3559-FSE-000045), supported by Norte Portugal Regional Operational Programme (NORTE 2020), under the PORTUGAL 2020 Partnership Agreement. ForestWISE—Collaborative Laboratory for Integrated Forest & Fire Management, was recognized as a CoLAB by the Foundation for Science and Technology, I.P. (FCT). https://www.forestwise.pt/pt/institution/

This article/publication is based upon work from COST Action, supported by COST (European Cooperation in Science and Technology). https://www.cost.eu/

The funders had no role in study design, data collection and analysis, decision to publish, or preparation of the manuscript.)

5. We note that Figure 1 in your submission contain map image and Figure 2 contain satellite image, which may be copyrighted. All PLOS content is published under the Creative Commons Attribution License (CC BY 4.0), which means that the manuscript, images, and Supporting Information files will be freely available online, and any third party is permitted to access, download, copy, distribute, and use these materials in any way, even commercially, with proper attribution. For these reasons, we cannot publish previously copyrighted maps or satellite images created using proprietary data, such as Google software (Google Maps, Street View, and Earth). For more information, see our copyright guidelines: http://journals.plos.org/plosone/s/licenses-and-copyright.

a) You may seek permission from the original copyright holder of Figure(s) [#] to publish the content specifically under the CC BY 4.0 license.  

Reviewers' comments:

Reviewer's Responses to Questions

**Comments to the Author**

1. Is the manuscript technically sound, and do the data support the conclusions?

Reviewer #1: Yes

Reviewer #2: Yes

2. Has the statistical analysis been performed appropriately and rigorously? 

Reviewer #1: Yes

Reviewer #2: Yes

3. Have the authors made all data underlying the findings in their manuscript fully available?

Reviewer #1: Yes

Reviewer #2: No

4. Is the manuscript presented in an intelligible fashion and written in standard English?

Reviewer #1: Yes

Reviewer #2: Yes

5. Review Comments to the Author

Reviewer #1: Please state clearly in the text:

1. How many individuals were the goat samples used? Whether the 185 muscle samples came from different individuals

2. How are these individuals related?

3. What are the reasons for choosing 14 microsatellite markers?

4. Please refer to the journal requirements to modify the table format.

5. FIG. 2, this labeling method is difficult to identify, so it is suggested to adjust the background color of the satellite map or replace it with a blank background containing latitude and longitude labels.

6. It is recommended to start with the introduction or discussion to simplify the article, which should correspond to the conclusion

Reviewer #2: I have evaluated the paper “Genetic signature of blind reintroductions of Iberian ibex (Capra pyrenaica) in Catalonia, Northeast Spain”. In this study, the authors characterize the variability of three Iberian ibex populations from Catalonia by using mitochondrial sequences and 14 microsatellite markers. No signature of introgression with domestic goats is identified and a close affinity between the Tortosa Beseit and Montserrat populations is detected, while the Montgrí population appears to be more differentiated. Conservation and management strategies are also discussed in the light of the results obtained by the authors. This paper can be of interest to readers of PloS One, but a major revision is needed. Please provide a detailed response to each one of the queries outline below indicating in which lines of the new manuscript changes have been made

With regard to the scientific content of the paper

Q1.My main observation is that a representative number of domestic goats must be genotyped for the 14 microsatellites and such data should be included in the Structure plot. Otherwise, it is hard to understand how the authors conclude that microsatellite data corroborate the absence of hybridization with domestic goats (?).An added benefit of genotyping some goats would be to compare their levels of diversity with those of ibexes. A PCA including the 3 ibex populations plus a domestic goat population would be also welcome.

Q2.In the Results section, please divide it in 2 subsections: one for mitochondrial data and another one for microsatellite data. In L315 indicate the FST value even if it is not significant. Sometimes it is difficult to infer whether you are talking about mitochondrial or microsatellite results. This needs to be presented in a much clearer way.

Q3.I have difficulties in understanding the “assignment test”. The basis of such analysis needs to be explained with more detail in the materials and methods section. In L307-8 is explained that the assignment test is based on Structure results and I assume that this refers to microsatellite genotypes, right? Then the clusters should be Tortosa Beseit+Montserrat and Montgrí (K=2), or Tortosa Beseit, Montserrat and Montgrí. So, how an assignment probability can be calculated for domestic goats in Table S3? Based on mitochondrial data? But mitochondrial data are haploid, so no point in making a Structure analysis. The authors are using an admixture model assuming that individual i has inherited some fraction of its genome from ancestors in population k and yielding, as output, the posterior mean estimates of these proportions. Please check whether the term is “posterior probablilities” used in the legend of Fig. 4 is correct in this context (in the non-admixture model it would, but I am not so sure about the admixture model so I advise to verify it). Data presented in Table S3, from where do they come from? What is genetic cluster 1 and 2? Cluster 2 is domestic goats? But not a single domestic goat has been genotyped with microsatellites and subsequently analysed with Structure, right? Each Table and Figure should be meaningful by itself, so please indicate with precision what are you displaying because this part of the text is very confusing. Indicate in a clear way which type of marker and which clusters or popultions are you taking into consideration in each of the analyses. Detailed legends for tables and figures are very helpful to readers. Detailed methodological explanations are also very welcome.

Q4.What is your interpretation about the failure to detect bottlenecks with the Bottleneck software despite the demographic history of these 3 ibex populations? Is the Bottleneck software well suited to detect any type of bottleneck? This should be discussed properly.

Q5.Does it make sense that the Montserrat population is non-inbred (see lines 453-4) and its source population (Tortosa Beseit) is inbred based on FIS measurements despite of having a much larger size? I find it very hard to believe (how a population recently derived from an inbred source population can be non-inbred?). This needs to be discussed. With 46 founders and no additional incoming individuals, it is impossible that the Montserrat population is not inbred. Maybe this is explained by the inherent lack of power of FIS to detect inbreeding when scarce molecular data are available. In any case, an interpretation should be provided to understand this unexpected outcome.

Q6.L135, 370, 371, 387. I am nost sure whether is realistic to hypothesize that Iberian ibexes might be introgressed with Alpine ibexes or vice versa. Their geographic ranges are completely different and they are separated by 1,000 km, so how could the introgression of Iberian ibexes with Alpine ibexes be a threat (L136)? Such hypothesis needs to be justified or discarded as unrealistic.

Q7.L193, sequencing protocol should be explained

Q8.L325, why a high HWE departure is expected? The numbers of microsatellites in HWD are quite low when populations are considered individually, and I am not sure if it is correct to merge three different populations into a single one and then state that 8/14 microsatellites were in disequilibrium. In other words, could population stratification contribute to this HW departure?

Q9.L404,is it correct to say that TBNGR population has a high diversity? If compared to domestic goats, sure that it does not. This population in 1966 had around 400-500 individuals.

Q10. Both microsatellite genotypes and mitochondrial sequences should be made available. Please include an availability statement in the manuscript.

With regard to formal issues:

Q11. There are many typos that should have been corrected before all authors approved the final submission of the paper.

In many places populations are named in (at least) 2 different manners

e.g Montserrat (correct) and Monserrat (incorrect, please ammend)

Cataluña and Catalonia (use Catalonia since it is an English paper)

Montgrí (correct) and Montgri (incorrect)

Tortosa y Beceit (Admixture plot, incorrect), Tortosa-Beseit (correct), Beseit (L451, incorrect), Tortosa i Beseit (correct), Tortosa and Beseit (correct). Although 3 terms are correct, this population needs to be named with just one single denomination, not multiple different names. Please pick one of these denominations and use it consistently throughout the paper.

The format of the references is also completely inconsistent, as if anybody had revised it. Even if a free-format is allowed, this does not imply than the references can be written in multiple inconsistent formats. Some examples: L572, 3-37.3. (??), plus minus sign in L587 and many other places, L584, a DOI is provided, but only for this ref. , L596,, lacks volume number and pages, L663, 1994 November, L664 PMID etc etc. Please use a free but coherent format to cite papers, and make sure that you are citing them correctly.

Other formal issues:

L138, Iberian peninsula

L174, none of the animals were hunted

L182-3, from the 185 muscle samples, please indicate the number of samples for each population (between parentheses).

L187, 2.5 microliters of 10x buffer

L188, please recheck whether dNTP concentrations is 10 micromolar. It seems quite low to me. MgCl2, the 2 should be subindexed

L199, electrophoresed, not sequenced (if you are talking about microsatellites)

L209, I would say diagnosis

L216, electropherograms

L217, visual inspection (not manual)

L247, indicate how many individuals from each population were successfully sequenced

L292, do you mean 96 ibexes with data for 14 microsatellites?

L308, STRUTUCTURE

L324, sensible means “of good judgement”. Replace by sensitive

L369, gene flow

L370, Alpine

L402, nuclear what?

L423,424, FST without subindexing ST

L451, matrilineal

L497, genetic diversity (not traits)

L508, not sure vein territories is correct, maybe neighboring?

The resolution of the network Figure is too low and the names of the populations are unreadable (please write them with a larger font).

In Table 2, what is the meaning of IAM = xxx and SMM = dxxx? Besides, abbreviations could be defined in a footnote rather than in the legend of the table.

Figure 2, needless to write gene pool 1, 2 and 3. Do not superimpose K values onto the Figure

Instead of Supporting materials I would say Supporting or Supplementary Tables. I see a list of references below Table S2, but I am not sure whether refs corresponding to Table S1 are mentioned anywhere (?).

Table S3, decimals separated with points, not commas

This list of errors/typos is not exhaustive, so I advise all authors to take a thorough look at the revised paper.

6. PLOS authors have the option to publish the peer review history of their article (what does this mean?). If published, this will include your full peer review and any attached files.

Reviewer #1: No

Reviewer #2: No

---

## [Author Response · Author response to Decision Letter 0]

13 Apr 2022

Manuscript number: PONE-D-21-37151

Title: Genetic signature of blind reintroductions of Iberian ibex (Capra pyrenaica) in Catalonia, Northeast Spain

Journal: PLOS ONE

RESPONSE TO REVIEWERS’ COMMENTS

We appreciate all the reviewers’ comments, and we thank them for their relevant contribution to the improvement of the manuscript. We have carefully read the reviewers’ remarks and explain below how we delt with each one. Overall, we believe the revision has improved the contents and clarity of the manuscript, and we hope that all the reviewers’ concerns have been fully addressed.

NOTE: Some references were deleted and others were added. The references are completely updated in the revised version of the manuscript with no track changes.

RESPONSE TO REVIEWER #1

Q1:

How many individuals were the goat sampled used? Whether the 185 muscle samples came from different individuals.

Our response: The muscle samples used for this study were collected from legally hunted individuals, therefore all 185 muscle samples came from different individuals. We changed the manuscript and discriminated samples by populations for clarity.

Q2:

How are these individuals related?

Our response: We thank you for your remark. We don’t have access to the nature of the relation between the sampled individuals. The available information that is provided in the manuscript, regarding the origin of Montserrat population (translocation of individuals from TBNGR) and Montgrí (5 runaway individuals from a zoo in Girona, that have been previously brought to this zoo from Andalusia). This information can be found in the manuscript in L122-125.

Q3:

What are the reasons for choosing 14 microsatellite markers?

Our response: We are aware that using a larger number of microsatellites would provide further strength to our analysis, but the number and diversity of markers was constrained by available resources (budget and time) and we aimed to optimize cost-benefit relation. We used the highest number of markers that was feasible given the available resources. After a thorough bibliographic research and with the aim of maximizing the obtained data, we selected these microsatellite markers due to i) their successful application in previous studies regarding caprine species and ii) their level of polymorphism, range and annealing temperature (we selected the most polymorphic markers, that were feasible to co-amplify without the risk of presenting overlapping allele ranges). The number of markers is within the range that is commonly used in similar studies (https://doi.org/10.1371/journal.pone.0231832, https://doi.org/10.1002/ece3.4872, https://doi.org/10.1371/journal.pone.0207662, https://doi.org/10.1371/journal.pone.0231832, https://doi.org/10.1371/journal.pone.0220746, https://doi.org/10.1371/journal.pone.0213515, https://doi.org/10.1371/journal.pone.0216549, and the microsatellite panel we used was enough to clearly discriminate between: (i) domestic and wild goats; (ii) and the goats from Montgri and those from TBNGR and Monserrat.

Q4:

Please refer to the journal requirements to modify the table format.

Our response: We thank you for your remark and corrected the manuscript according to the suggestion.

Q5:

FIG. 2, this labelling method is difficult to identify, so it is suggested to adjust the background colour of the satellite map or replace it with a blank background containing latitude and longitude labels.

Our response: We thank you for your remark and corrected the figure accordingly, in order to make it clearer.

Q6:

It is recommended to start with the introduction or discussion to simplify the article, which should correspond to the conclusion.

Our response: Unfortunately, we didn’t comprehend this question. Could you please clarify?

RESPONSE TO REVIEWER #2

Q1:

My main observation is that a representative number of domestic goats must be genotyped for the 14 microsatellites and such data should be included in the Structure plot. Otherwise, it is hard to understand how the authors conclude that microsatellite data corroborate the absence of hybridization with domestic goats (?). An added benefit of genotyping some goats would be to compare their levels of diversity with those of ibexes. A PCA including the 3 ibex populations plus a domestic goat population would be also welcome.

Our response: We agree with the reviewer and, in fact, 18 domestic goat samples had been originally genotyped as well, for the 14 microsatellites. The results of the assignment of domestic goats and Iberian wild goats, inferred using STRUCTURE, was presented in Supporting Material 3, in the original submission, but we fail to explain it clearly in the first version of the paper. We have now included these results also in the plot with individual posterior probability of assignment (actually now Q proportion of genotype assigned to each cluster), inferred by STRUCTURE, provided in Figure 4. As suggested, we also included an ordination diagram (PCoA) depicting the genetic relations among the three Iberian wild goat populations plus the domestic goat population.

Q2:

In the Results section, please divide it in 2 subsections: one for mitochondrial data and another one for microsatellite data. In L315 indicate the FST value even if it is not significant. Sometimes it is difficult to infer whether you are talking about mitochondrial or microsatellite results. This needs to be presented in a much clearer way.

Our response: We thank you for your remark and have altered the manuscript accordingly, dividing the Results section into mitochondrial data and microsatellite data, and added the requested FST value.

Q3:

I have difficulties in understanding the “assignment test”. The basis of such analysis needs to be explained with more detail in the materials and methods section. In L307-8 is explained that the assignment test is based on Structure results and I assume that this refers to microsatellite genotypes, right? Then the clusters should be Tortosa Beseit+Montserrat and Montgrí (K=2), or Tortosa Beseit, Montserrat and Montgrí. So, how an assignment probability can be calculated for domestic goats in Table S3? Based on mitochondrial data? But mitochondrial data are haploid, so no point in making a Structure analysis. The authors are using an admixture model assuming that individual i has inherited some fraction of its genome from ancestors in population k and yielding, as output, the posterior mean estimates of these proportions. Please check whether the term is “posterior probabilities” used in the legend of Fig. 4 is correct in this context (in the non-admixture model it would, but I am not so sure about the admixture model so I advise to verify it). Data presented in Table S3, from where do they come from? What is genetic cluster 1 and 2? Cluster 2 is domestic goats? But not a single domestic goat has been genotyped with microsatellites and subsequently analysed with Structure, right? Each Table and Figure should be meaningful by itself, so please indicate with precision what are you displaying because this part of the text is very confusing. Indicate in a clear way which type of marker and which clusters or populations are you taking into consideration in each of the analyses. Detailed legends for tables and figures are very helpful to readers. Detailed methodological explanations are also very welcome.

Our response: We thank you for your comments and agree that the genotyping of domestic goat samples and comparison with wild goats genotypes was not clearly described in the paper. All STRUCTURE runs were performed using genotypic polymorphic (microsatellite) data and not on mitochondrial data. STRUCTURE is based on a clustering method to infer population structure and assign individuals to populations through genotypic data, using a population (rather than individual) optimization criterium. However, the reviewer is correct. According to Pritchard et al., 2000, in the non-admixture model, the parameter that is estimated for each individual is the “posterior probability of assignment of the individual to each population”, the correct interpretation for the admixture model is “the proportion of the genome of the individual that is assigned to each population”. We have now corrected this in the paper, in all the tables, figures and supplementary information.

Last, for clarifying, all the individuals, both wild and domestic, were genotyped in this study. With that in mind, 17 domestic goats were genotyped with 14 microsatellites and subsequently analysed with STRUCTURE. We made the necessary manuscript changes to make that clear and included these Structure results in Figure 4.

Q4:

What is your interpretation about the failure to detect bottlenecks with the Bottleneck software despite the demographic history of these 3 ibex populations? Is the Bottleneck software well suited to detect any type of bottleneck? This should be discussed properly.

Our response: We thank the reviewer by the pertinent comment. We agree that the method described by Cornuet & Luikart (1996) is less powerful than alternative methods in detecting the sign of recent bottlenecks, for the transient increase in heterozygosity is rapidly eroded after a few generations after the bottleneck event. We more frequently use the method described by Garza & Williamson (2001), which targets the relation between the number of alleles and the allele size range of an allele distribution, which is expected to retain the signal of the bottleneck for a longer period. We used to calculate these values using the executable files created by the authors of the method (Garza & Williamson, 2001), that worked on Mac OS Classic operative system, and allowed for the estimation of the significance of the estimated M values, in different simulated scenarios. Unfortunately, we have no longer access to such executable files and were not able to estimate these values and associated significance.

In response to the reviewer’s comment, we have searched for alternative ways of estimating this ratio, that in our view, is more informative. We found out that ARLEQUIN package does estimate the values, unfortunately without estimating the associated significance. However, Garza & Williamson (2001), in their description of the method have detected a theoretical threshold, of about two thirds (0.68) that would be an indicative value below which there would be strong evidence of bottlenecked populations. We replaced these values of the M ratio and discuss them in light of this theoretical threshold. In doing this estimation of M ratio, we excluded two markers (BM1258, IDVGA30) that had much more missing data than the remaining twelve markers. Given the fact that the ratio is highly sensitive to the number of different alleles, we fear that using markers with a large amount of missing data would artificially lower the ratio and was not a conservative approach. So we estimated the M ratio using all other 12 markers and report this value in the paper (we actually tested both approaches and results were as expected). Nevertheless, the estimated values using only 12 markers are still very close to the theoretical threshold estimated by Garza & Williamson (2001), therefore suggesting the occurrence of recent bottlenecks, which is consistent with the known demographic history of these populations. According to these authors, with as few as 8 markers, would be theoretically possible to detect bottleneck events.

Reference: Garza, J.C.; Williamson, E.G. Detection of reduction in population size using data from microsatellite loci. Mol. Ecol. 2001, 10, 305–318

Q5:

Does it make sense that the Montserrat population is non-inbred (see lines 453-4) and its source population (Tortosa Beseit) is inbred based on FIS measurements despite of having a much larger size? I find it very hard to believe (how a population recently derived from an inbred source population can be non-inbred?). This needs to be discussed. With 46 founders and no additional incoming individuals, it is impossible that the Montserrat population is not inbred. Maybe this is explained by the inherent lack of power of FIS to detect inbreeding when scarce molecular data are available. In any case, an interpretation should be provided to understand this unexpected outcome.

Our response: Genetic isolation is not absolute among the three populations, in particular in the case of Tortosa I Beceit and Monserrat. In fact, it is likely that individuals could disperse from an area to the other. Population size in Tortosa I Beceit is larger and it is less likely that a limited number of individuals coming from other populations might dramatically affect diversity indices and genetic parameters of this population. However, the population of Monserrat is smaller and resulted from a founder event with a limited number of individuals. In a smaller population, with a small number of genotyped individuals and scarce molecular data (as the reviewer has stressed), the effect of incoming migrants might have more marked effects on genetic parameters. Namely, the sampling of newly arrived migrants with very distinct gene pool, or its descendants, may mask the effects of inbreeding, as a result of outbreeding of the new migrants with individuals that descend from the founder gene pool. Results from the STRUCUTRE runs confirm that there is, in fact, at least one individual from Monserrat that is assigned to a distinct gene pool.

Q6:

L135, 370, 371, 387. I am nost sure whether is realistic to hypothesize that Iberian ibexes might be introgressed with Alpine ibexes or vice versa. Their geographic ranges are completely different and they are separated by 1,000 km, so how could the introgression of Iberian ibexes with Alpine ibexes be a threat (L136)? Such hypothesis needs to be justified or discarded as unrealistic.

Our response: The reviewer is correct, in the sense that it is not reasonable to assume natural introgression of these two species, given the disjunct geographic ranges. However, when we addressed this issue, we were not focusing on naturally occurring introgression but on human-mediated processes, such as admixture resulting from translocations. Namely, in the Girona zoo, different ungulate and even Capra species are herded together, so introgression could have occurred there. While not very likely to occur, we consider that is a threat that was relevant to assess. We have modified the introduction and discussion in order to clarify our hypothesis.

Q7:

L193, sequencing protocol should be explained

Our response: We described the sequencing protocol in general terms. We hope that the information is enough and are available for additional clarifications. PCR products were visualized in agarose gels, purified and sequenced (Sanger) in an automatic sequencer ABIPRISM® 3730-XL DNA Analyzer, Applied Biosystems™, using BigDye™ sequencing kit

Q8:

L325, why a high HWE departure is expected? The numbers of microsatellites in HWD are quite low when populations are considered individually, and I am not sure if it is correct to merge three different populations into a single one and then state that 8/14 microsatellites were in disequilibrium. In other words, could population stratification contribute to this HW departure?

Our response: Thank you for your remarks. One of the main causes of departure from HWE and presence of LD is the presence of structure within a population. So yes, presence of population structure (evidenced through our Structure results) is the most likely cause of such deviations. In fact, this was our aim when addressing this question. We aimed to reinforce the structured nature of this population, by calling attention to the deviations to HWE, that are in line with all the other evidences of population structure.

Q9:

L404, is it correct to say that TBNGR population has a high diversity? If compared to domestic goats, sure that it does not. This population in 1966 had around 400-500 individuals.

Our response: When we state that TBNGR is a population with high diversity, we are necessarily referring to natural populations and we consider that we cannot compare with the diversity in domestic goats. Natural populations are normally more isolated geographically and dependent on natural dispersal than domestic populations. With exception to human-mediated translocations and events of natural dispersal natural populations are more dependent on the local or regional gene pool. On the other hand, domestic breeds are often artificially selected in different parts of the world and interbred with other breeds with very distinct gene pools (namely from European or Asian origin, such as in the case of pig or dog). In fact, its is often the case that the local populations of a wild species present less diversity than the “locally sampled populations” of domestic conspecifics. This is true for pigs (Alexandri et al, 2012) or dogs (Vila et al, 1997; Godinho et al, 2011; Torres et al, 2017) and could also be expected for goats. To be able to assess the level of diversity of the TBNGR population we must compare with other natural populations.

Q10:

Both microsatellite genotypes and mitochondrial sequences should be made available. Please include an availability statement in the manuscript.

Our response: The haplotypes will be submitted to Genbank immediately after the acceptance for publication, so accession numbers are available for being included in the paper, prior to final acceptance and paper editing. Microsatellite genotypes will be provided after the acceptance for publication. 

Q11:

There are many typos that should have been corrected before all authors approved the final submission of the paper. In many places populations are named in (at least) 2 different manners e.g Montserrat (correct) and Monserrat (incorrect, please ammend), Cataluña and Catalonia (use Catalonia since it is an English paper), Montgrí (correct) and Montgri (incorrect), Tortosa y Beceit (Admixture plot, incorrect), Tortosa-Beseit (correct), Beseit (L451, incorrect), Tortosa i Beseit (correct), Tortosa and Beseit (correct). Although 3 terms are correct, this population needs to be named with just one single denomination, not multiple different names. Please pick one of these denominations and use it consistently throughout the paper. The format of the references is also completely inconsistent, as if anybody had revised it. Even if a free-format is allowed, this does not imply than the references can be written in multiple inconsistent formats. Some examples: L572, 3-37.3. (??), plus minus sign in L587 and many other places, L584, a DOI is provided, but only for this ref., L596,, lacks volume number and pages, L663, 1994 November, L664 PMID etc etc. Please use a free but coherent format to cite papers, and make sure that you are citing them correctly.

Our response: We thank you for your remarks, which we followed and changed the manuscript accordingly. We have revised the populations denominations and standardize them, using only one denomination for each. We have also revised and corrected all the references and cited them correctly.

Other formal issues:

Reviewer #2:

L138, Iberian Peninsula

Our response: We thank you for your suggestion, which we followed and changed the manuscript accordingly.

Reviewer #2:

L174, none of the animals were hunted.

Our response: We thank you for your remark and rephrased the sentence according to the suggestion.

Reviewer #2:

L182-3, from the 185 muscle samples, please indicate the number of samples for each population (between parentheses).

Our response: We agree with the reviewer and therefore discriminated the number of samples for each population in the manuscript.

Reviewer #2:

L187, 2.5 microliters of 10x buffer. L188, please recheck whether dNTP concentrations is 10 micromolar. It seems quite low to me. MgCl2, the 2 should be subindexed.

Our response: Thank you for your remarks. The reviewer is correct. We used 2.5microliters of 10x buffer and the correct dNTP concentration is 10 milimolar. We corrected both, but preferred to refer to 25ul of final volume with 1x reaction buffer. We also corrected the subscript in MgCl2.

Reviewer #2:

L199, electrophoresed, not sequenced (if you are talking about microsatellites).

Our response: We thank you for your comment and changed “sequenced” for “genotyped”, which is a broader concept than electrophoresed, for it refers to the amplification, electrophoresis, bioinformatic analysis, allele calling and genotyping.

Reviewer #2:

L209, I would say diagnosis.

Our response: We agree with the reviewer and made the necessary manuscript changes.

Reviewer #2:

L216, electropherograms.

Our response: We thank you for your remark and corrected the manuscript according to the suggestion.

Reviewer #2:

L217, visual inspection (not manual).

Our response: We thank you for your correction and changed the manuscript accordingly.

Reviewer #2:

L247, indicate how many individuals from each population were successfully sequenced.

Our response: We thank you for your suggestion and changed the manuscript accordingly, discriminating how many samples were successfully sequenced for each population.

Reviewer #2:

L292, do you mean 96 ibexes with data for 14 microsatellites?

Our response: Yes, these 96 individual genotypes correspond to 96 individual ibexes with data for the selected 14 microsatellites.

Reviewer #2:

L308, STRUTUCTURE.

Our response: We thank you for your correction and changed the manuscript accordingly.

Reviewer #2:

L324, sensible means “of good judgement”. Replace by sensitive.

Our response: We thank you for your remark and corrected the manuscript according to the suggestion.

Reviewer #2:

L369, gene flow.

Our response: We thank you for your correction and changed the manuscript accordingly.

Reviewer #2:

L370, Alpine.

Our response: We thank you for your correction and changed the manuscript accordingly.

Reviewer #2:

L402, nuclear what?

Our response: We agreed with the reviewer that the information was not clear and corrected the manuscript accordingly.

Reviewer #2:

L423,424, FST without subindexing ST.

Our response: We thank you for your correction and changed the manuscript accordingly.

Reviewer #2:

L451, matrilineal.

Our response: We thank you for your correction and changed the manuscript accordingly.

Reviewer #2:

L497, genetic diversity (not traits).

Our response: We thank you for your remark and corrected the manuscript according to the suggestion.

Reviewer #2:

L508, not sure vein territories is correct, maybe neighbouring?

Our response: We thank you for your correction and changed the manuscript accordingly.

Reviewer #2:

The resolution of the network Figure is too low and the names of the populations are unreadable (please write them with a larger font).

Our response: We thank you for your suggestion and revised the figure accordingly, writing the population names with a larger font.

Reviewer #2:

In Table 2, what is the meaning of IAM = xxx and SMM = dxxx? Besides, abbreviations could be defined in a footnote rather than in the legend of the table.

Our response: We changed this section, as we tested bottlenecks through another method. Analysis, results, and discussion were rearranged in the revised version of the manuscript.

Reviewer #2:

Figure 2, needless to write gene pool 1, 2 and 3. Do not superimpose K values onto the Figure. 

Our response: We thank you for your suggestion and have revised the figure accordingly.

Reviewer #2:

Instead of Supporting materials I would say Supporting or Supplementary

Our response: We thank you for your remark. The use of Supporting materials is in accordance with the journal guidelines.

Reviewer #2:

Tables. I see a list of references below Table S2, but I am not sure whether refs corresponding to Table S1 are mentioned anywhere (?).

Our response: We thank you for your remark and added a list of references below Table S1.

Reviewer #2:

Table S3, decimals separated with points, not commas. This list of errors/typos is not exhaustive, so I advise all authors to take a thorough look at the revised paper.

Our response: We thank you for your correction and changed the table accordingly.

---

## [Decision Letter · Decision Letter 1]

30 May 2022

Genetic signature of blind reintroductions of Iberian ibex (Capra pyrenaica) in Catalonia, Northeast Spain

PONE-D-21-37151R1

Dear Dr. Barros,

We’re pleased to inform you that your manuscript has been judged scientifically suitable for publication and will be formally accepted for publication once it meets all outstanding technical requirements.

Kind regards,

Tzen-Yuh Chiang

Academic Editor

PLOS ONE

Additional Editor Comments (optional):

Reviewers' comments:

Reviewer's Responses to Questions

**Comments to the Author**

1. If the authors have adequately addressed your comments raised in a previous round of review and you feel that this manuscript is now acceptable for publication, you may indicate that here to bypass the “Comments to the Author” section, enter your conflict of interest statement in the “Confidential to Editor” section, and submit your "Accept" recommendation.

Reviewer #2: All comments have been addressed

2. Is the manuscript technically sound, and do the data support the conclusions?

Reviewer #2: Yes

3. Has the statistical analysis been performed appropriately and rigorously? 

Reviewer #2: Yes

4. Have the authors made all data underlying the findings in their manuscript fully available?

Reviewer #2: No

5. Is the manuscript presented in an intelligible fashion and written in standard English?

Reviewer #2: Yes

6. Review Comments to the Author

Reviewer #2: The authors have addressed my queries correctly. I have a few amendments that the authors might take into consideration:

L123, that have

L149, at stake

L260, Fig. 3?

L377, MonTserrat

L441-4, I would rewrite this sentence (this population appears 4 times and a populaion carried out by few individuals sounds awkward)

L520, might occur

L542, Agents Rurals corp

7. PLOS authors have the option to publish the peer review history of their article (what does this mean?). If published, this will include your full peer review and any attached files.

Reviewer #2: No

---

## [Editor Report · Acceptance letter]

9 Sep 2022

PONE-D-21-37151R1 

Genetic signature of blind reintroductions of Iberian ibex (*Capra pyrenaica*) in Catalonia, Northeast Spain 

Dear Dr. Barros:

I'm pleased to inform you that your manuscript has been deemed suitable for publication in PLOS ONE. Congratulations! Your manuscript is now with our production department. 

Kind regards, 

on behalf of

Dr. Tzen-Yuh Chiang 

Academic Editor

PLOS ONE